# Alternative Lengthening of Telomeres and Mediated Telomere Synthesis

**DOI:** 10.3390/cancers14092194

**Published:** 2022-04-27

**Authors:** Kailong Hou, Yuyang Yu, Duda Li, Yanduo Zhang, Ke Zhang, Jinkai Tong, Kunxian Yang, Shuting Jia

**Affiliations:** 1Faculty of Life Science and Technology, Kunming University of Science and Technology, 727 Jing Ming Nan Road, Kunming 650500, China; 20213136001@stu.kust.edu.cn; 2Laboratory of Molecular Genetics of Aging and Tumor, Medical School, Kunming University of Science and Technology, 727 Jing Ming Nan Road, Kunming 650500, China; 20192136002@stu.kust.edu.cn (Y.Y.); 20192136015@stu.kust.edu.cn (D.L.); 20212136016@stu.kust.edu.cn (Y.Z.); 20202136022@stu.kust.edu.cn (K.Z.); 20212136004@stu.kust.edu.cn (J.T.); 3First People’s Hospital of Yunnan Province, 157 Jinbi Road, Kunming 650032, China

**Keywords:** telomere maintenance mechanisms, alternative lengthening of telomeres, homologous recombination

## Abstract

**Simple Summary:**

Alternative lengthing of telomere (ALT) is an important mechanism for maintaining telomere length and cell proliferation in telomerase-negative tumor cells. However, the molecular mechanism of ALT is still poorly understood. ALT occurs in a wide range of tumor types and usually associated with a worse clinical consequence. Here, we review the recent findings of ALT mechanisms, which promise ALT could be a valuable drug target for clinical telomerase-negative tumor treatment.

**Abstract:**

Telomeres are DNA–protein complexes that protect eukaryotic chromosome ends from being erroneously repaired by the DNA damage repair system, and the length of telomeres indicates the replicative potential of the cell. Telomeres shorten during each division of the cell, resulting in telomeric damage and replicative senescence. Tumor cells tend to ensure cell proliferation potential and genomic stability by activating telomere maintenance mechanisms (TMMs) for telomere lengthening. The alternative lengthening of telomeres (ALT) pathway is the most frequently activated TMM in tumors of mesenchymal and neuroepithelial origin, and ALT also frequently occurs during experimental cellular immortalization of mesenchymal cells. ALT is a process that relies on homologous recombination (HR) to elongate telomeres. However, some processes in the ALT mechanism remain poorly understood. Here, we review the most recent understanding of ALT mechanisms and processes, which may help us to better understand how the ALT pathway is activated in cancer cells and determine the potential therapeutic targets in ALT pathway-stabilized tumors.

## 1. Introduction

Telomeres are located at the end of eukaryotic chromosomes, and in humans, they are composed of TTAGGG tandem repeat DNA sequences and telomere-binding proteins [1]. They are special structures that do not carry genetic information, and they comprise a proximal double-stranded region and the distal single-stranded region [2]. Telomeres prevent the loss of genetic information during DNA replication and protect chromosomes from end fusion. Except in embryonic germ cells, stem cells, and cancer cells, telomere length gradually shortens with cell division [3]. Short or dysfunctional telomeres are recognized as double-strand breaks (DSBs), triggering replicative senescence of cells [4].

Telomere maintenance is essential for genomic stability and survival of proliferating cells. To escape from the “Hayflick limit”, the majority of tumor cells reactivate telomerase, which maintains telomere length [5,6,7]. Telomerase is a ribonucleoprotein polymerase that maintains telomere length by adding telomere DNA repeats to the 3′-OH end of telomeres [8]. This enzyme consists of a protein component with reverse transcriptase activity and an RNA component that is the template for telomeric DNA synthesis [9] (Figure 1). However, approximately 10 to 15% of human tumors preferentially maintain telomeres through the alternative lengthening of telomeres (ALT) pathway (Figure 1), which is a potential therapeutic target for telomerase-negative tumors [10,11]. The ALT phenotype has been observed in a broad range of human cancers, and some ALT-related cancers are aggressive [12,13]. However, the development of anticancer therapeutics targeting the ALT pathway has been greatly limited by a failure to understand the molecular mechanisms underlying ALT pathway action and initiation. Here, we review recent discoveries regarding the ALT pathway mechanism and discuss possible cancer therapy targets in the ALT pathway.

## 2. ALT and ALT-Related Phenotypes

Recent research has suggested that ALT pathway is a telomere maintenance mechanism that depends on telomere sequence-specific homologous recombination (HR) [14]. The telomeres in ALT cells are lengthened by the recombination-mediated synthesis using the DNA sequences of other telomeres as templates, offsetting the shortening of telomeres caused by continuous cell division as well as oxidative stress [9]. Studies have shown that ALT pathway activation is typically associated with a characteristic phenotype, including highly heterogeneous cells, heterogenous telomere length, telomeric sister chromatid exchanges (T-SCEs), abundant extrachromosomal telomeric repeats (ECTRs), and nuclei containing ALT pathway-associated promyelocytic leukemia (PML) bodies (APBs) [11].

### 2.1. ALT-Associated PML Bodies

Among ALT pathway-related characteristic phenotypes, APBs are among the most representative [15]. APBs are observed in approximately 90% of ALT pathway-positive (ALT+) cells and are formed by the PML and Sp100 shell, bound together by SUMO-interacting motif (SIM) interactions and other proteins involved in DNA repair, recombination, replication, as well as telomeric DNA and shelterin proteins [16,17]. Although the function of APBs is not entirely clear, it has been hypothesized that APBs contribute to ALT pathway-based telomere maintenance. Some findings have indicated that APBs are active centers in the ALT pathway that can promote telomere clustering and serve as platforms to concentrate proteins, such as RPA, Rad51, BLM, and BRCA1, that are required for telomere maintenance through ALT [18,19]. Indeed, ALT+ cells lost key ALT hallmarks and underwent progressive telomere shortening when APB formation was disrupted by PML depletion [20].

Moreover, SUMOylation of APB components is essential for APB formation during the ALT process. Many telomere proteins are SUMOylated and/or contain SIMs [16,18,21]. SUMO-SIM interactions drive APB liquid condensation, which causes telomere clustering and membraneless organelle formation through liquid-liquid phase separation (LLPS) [22]. This promotes telomere DNA synthesis in the ALT pathway by concentrating the substrates and enzymes required for recombination-based telomere elongation. However, APBs are likely not critical in the ALT pathway. Some ALT+ cells have been reported to maintain telomeres in the absence of APBs [23,24]. Taylor et al. recently showed that recruitment of the BTR complex (containing BLM, TOP3α, RMI1, and RMI2) to telomeres is sufficient to induce the acquisition of ALT-related phenotypes in a PML-independent manner [20].

### 2.2. Extrachromosomal Telomeric Repeats

The ALT pathway is characterized by a high extrachromosomal telomeric repeat number, which includes a single-stranded C-rich circle and a double-stranded telomere circle (T-circle). The abundance of C-circles correlates with the DNA synthesis of telomeres in ALT+ cells; a greater C-circle abundance indicates higher ALT pathway activity [25,26]. However, how the C-circle is produced in ALT+ cells is unknown. Studies have shown that telomere DNA damage, especially DSBs, promotes C-circle production in ALT+ cells, and deletion of telomere replication-related proteins such as SMARCAL1 or CTC1-STN1-TEN1 (CST) can increase C-circle abundance in ALT+ cells [27,28]. This finding suggests that C-circle formation is related to telomeric DNA damage repair or replication defects in ALT+ cells. The T-loop is formed by the invasion of a single-strand terminus into an inner telomere strand, which inhibits replication because the strands need to be unwrapped to permit replication fork movement through the telomeres. It has been proposed that T-circles are formed either by intramolecular recombination between repeat sequences within telomeres or by the excision of a sequence from the T-loops [29]. In ALT+ cells, the T-circle can serve as a template for telomere elongation through rolling-circle replication. Aberrant HR of telomere sequences leads to the recruitment of the DNA repair proteins, XRCC3, NBS1, and SLX4, causing T-loop excision in the context of the ALT pathway [30,31]. Moreover, FANCD2 depletion and overexpression of the helicase BLM leads to an increase in ECTRs in ALT+ cells, revealing that ECTR DNA is associated with the intramolecular resolution of stalled replication forks in telomeric DNA [32,33]. In addition, it has recently been reported that damaged single-strand telomeres tend to form internal loops (I-loops), which comprise the majority of extrachromosomal telomeric circle structures detected in the ALT cells and function as substrates for the generation of extrachromosomal telomeric circles, such as C-circles [34].

### 2.3. Telomere Sister Chromatid Exchange (T-SCE)

ALT+ cells exhibit a high frequency of T-SCEs [35]. Unequal T-SCEs can result in deleterious consequences: some daughter cells inherit shorter telomeres and suffer telomere loss, while other daughter cells have longer telomeres and, therefore, have a prolonged proliferation capacity [36]. Although the molecular mechanism of T-SCE remains unknown, it has been suggested that shelterin cooperates with the Ku complex to inhibit T-SCEs. In the absence of the Ku70/80 complex, TRF2, RAP1, or POT1 deletion stimulates exchanges between telomeres on sister chromatids [31,37,38]. Although T-SCE is a widely used marker for ALT pathway activity, studies have shown that T-SCE is unlikely to be the mechanism of telomere maintenance in ALT+ cells [39]. Therefore, T-SCEs seem to contribute to telomere length heterogeneity in ALT+ cells, but other types of recombination-based events must also play a role in ALT-based telomere maintenance.

### 2.4. Chromatin Remodeling and Gene Mutations

Several recent studies have shown the importance of the telomere chromatin state in ALT processes. The chromatin factor alpha-thalassemia/mental retardation X-linked chromatin remodeler (ATRX), death domain-associated protein (DAXX), and H3.3 undergo high-frequency mutations in ALT+ tumor cells [40,41]. Located on the X chromosome, ATRX is the causative gene of alpha thalassemia/mental retardation syndrome. The ATRX gene encodes a specific ATRX protein that belongs to the SWI/SNF2 family of chromatin-remodeling proteins [41]. ATRX proteins are mainly enriched in repeat sequences of telomeres, subtelomeres, and pericentromeres. ATRX is a chromatin remodeler that partners with DAXX to generate the histone variant H3.3 and promote the cohesion of sister telomeres, which increases the break-induced replication (BIR) rate in ALT telomeres [41,42]. The ATRX/DAXX complex is critical for the assembly of histone variant H3.3 at telomeres, pericentromeres, and nucleosomal DNA repeat sequences [41]. The clinical data show deletions or mutations of ATRX in 95% of human ALT pathway-related tumor cells [43]. When ATRX is unexpressed in ALT cells, heterochromatin protein 1 (HP1α) is recruited to the telomeres, leading to telomere heterochromatinization, promoting the transcription of telomeric repeats containing RNA (TERRAs) in telomeres. A TERRA binds to telomeres to form R-loops, promoting ALT pathway activation [44]. The reintroduction of ATRX to ALT+ cells inhibits T-SCE, the formation of APBs and C-circles, and interchromosomal telomere recombination [45,46]. Moreover, knocking out ASF1a/b histone molecular chaperone expression may induce ALT pathway activity [47]. These chromatin changes have alluded to the initiation of HR.

### 2.5. Heterochromatin at Telomeres

Telomeric heterochromatin has been thought to suppress ALT pathway activity because highly condensed heterochromatin prevents recombination. However, some studies have shown that telomere heterochromatinization can promote ALT pathway activation [14,44,48].

Telomeres are heterochromatin-enriched regions, and the maintenance of heterochromatin requires the trimethylation of the lysine 9 residue of histone 3 (H3K9me3) [44]. The deletion of ATRX in ALT+ cells increases the abundance of the H3K9me3 modification on telomeres and stabilizes the dense telomeric chromatin structure [49]. The maintenance of heterochromatin at telomeres has been realized by the SETDB1-mediated deposition of the H3K9me3 mark at telomeres [44]. Moreover, HP1α overexpression increased the H3K9me3 mark abundance at telomeres [50]. Although telomeres exhibit a distinct heterochromatin structure, telomeres can be transcribed [51]. Telomere transcription starts at the promoter in a subtelomeric region and ends at the telomeric region, and the transcription product is called a telomeric repeat containing RNA (TERRA) [52], which can bind to telomeres and form an R-loop. According to a study on telomeric heterochromatin, telomeric heterochromatinization in ALT+ cells enhances telomeric transcriptional activity, producing more R-loops and promoting ALT pathway activity [44].

A recent study showed that TRIM28, KRAB-associated protein 1 (KAP1), and transcription intermediary factor 1β can bind to telomeres and suppress the activation of the ALT pathway [53]. In a subsequent study, TRIM28 was shown to stabilize SETDB1 and act as a scaffolding protein to recruit SETDB1 and HP1α, augmenting the H3K9me3 deposition at telomeres, thus promoting telomere heterochromatinization and suppressing ALT pathway activation [54]. These results are inconsistent with those of previous studies, and these controversial findings suggest that the mechanisms of ALT pathway activation require further investigation.

## 3. SMARCAL1 and FA Proteins Balance Replication Stress of ALT Telomeres to Ensure Telomere Synthesis

As a difficult-to-replicate DNA sequence, a telomeric repeat may not be readily replicated as exposing telomeres to many sources of stress can lead to DNA damage and replication defects. Compared with euchromatin regions, telomeres contain many repeated sequences, heterochromatin properties, and more complex secondary structures such as T-loops, R-loops, and G-quadruplexes [55]. As a result, telomere regions easily accumulate replication pressure when secondary telomere-related structures are not effectively removed during replication fork progression [56]. Studies have shown that ALT telomeres are prone to chronic replication stress; however, once replication stress exceeds a critical point, ALT+ cells die. Therefore, the mechanisms associated with replication stress solutions are required to ensure complete and accurate DNA synthesis in ALT+ cells. It has been shown that SMARCAL1 and FA proteins function to alleviate replication stress at ALT telomeres by promoting replication fork regression and restart, as well as by controlling ALT pathway activity to maintain a certain stress level and thus prevent ALT+ cell death [28,57].

### 3.1. SMARCAL1

SMARCAL1 is a replication fork remodeling enzyme that protects stalled forks from collapsing by promoting branch migration and fork regression [58]. SMARCAL1 can regulate the regression of stalled replication forks by annealing the nascent DNA strand and generating a four-way DNA structure that is similar to a Holliday junction (HJ) [58,59,60] (Figure 2A). SMARCAL1 is enriched at telomeres and resolves telomeric replication stress in ALT+ cells [28]. Previous studies have shown that deletion of SMARCAL1 led to stalled replication forks at ALT telomeres, which were then subject to cleavage by the structure-specific endonuclease SMX (SLX1-SLX4, MUS81-EME1, and XPF-ERCC1) complex [28]. Feng et al. demonstrated that deletion of SMARCAL1 promoted CSB protein localization on ALT telomeres and that the CSB protein promoted the recruitment of repair-related proteins such as RAD50, BLM, and POLD3 to ALT telomeres [61]. Moreover, CSB inhibited the recruitment of MUS81 and SLX4 to telomeres [62,63]. Deficiency of SMARCAL1 and/or CSB proteins not only increased the recruitment of the SMX complex to telomeres but also led to the formation of fragile telomeres [61]. This evidence suggests that the SMARCAL and CSB proteins cooperate to protect stalled replication forks from causing fragile telomere formation in ALT+ cells.

### 3.2. FA Proteins

Fanconi (FA) proteins constitute a family of highly conserved proteins that includes FANCM, FANCD2, FANCD1, and FANCF, among other proteins, which participate in the cellular DNA damage repair pathways [64]. The main FA proteins that participate in telomeric DNA damage repair in ALT+ cells are FANCM and FANCD2. BRCA1/2 and FANCD2 localize to stalled replication forks, protect nascent DNA strands from nuclease degradation, and restart replication forks after the repair is completed [65]. FANCD2 can inhibit BLM localization to damaged ALT telomeres, thereby protecting and restarting the replication fork [32]. The deletion of FANCM in ALT+ cells was shown to lead to telomere dysfunction and, with cell proliferation inhibited, to cell death [66]. Moreover, FANCM can unwind telomeric R-loops and suppress their accumulation in ALT+ cells, and the reduction in R-loops attenuates the formation of APBs and C-circles in ALT+ cells [67]. Robert Lu et al. also demonstrated that FANCM and BTR complexes inhibited ALT pathway activity [66]. Pan et al. demonstrated that the HR-related proteins, BRCA1 and BLM, are recruited to the damage site in FANCM-deficient cells and resolve telomeric replication stress by promoting DNA end resection and HR [68] (Figure 2A).

## 4. Initiation of the ALT Pathway: DNA Damage at ALT Telomeres

When a stalled replication fork fails to restart, the replication fork collapses, resulting in a DSB. ALT has long been considered an HR-dependent telomeric synthesis mechanism induced by DSBs [13]. However, ALT has been recently shown to resemble BIR in mammalian cells [69,70]. Recent studies have shown that transient DNA damage caused by an inducible TRF1-FokI system can lead to the recruitment of specific replicators, including proliferating cell nuclear antigen (PCNA), replication factor C (RFC), and DNA polymerase δ, which are necessary factors for break-induced telomere synthesis [69]. Polδ is related to ALT telomere replication, which suggests that break-induced telomere synthesis is essential for ALT pathway activity.

The activation of ALT requires an increased replication pressure, leading to the accumulation of a certain level of DSBs, which activates the ALT pathway. ALT+ cells exhibit more DNA replication problems at telomeres than human telomerase+ cells when ALT cells are generated by the CRISPR/Cas9 knockout of telomerase RNA (TERC) [70]. Studies have shown that the ALT machinery is activated by the DSBs generated by a stalled replication fork [69]. Therefore, ALT telomeres maintain a delicate balance between damage and repair, and disruption of this balance leads to ALT pathway dysregulation [66].

When an irreversibly stalled replication fork collapses, a single-ended DSB is generated, and this DSB is preferentially repaired by BIR. BIR during mitosis (also known as mitotic DNA synthesis, MiDAS) has been reported to be critical for telomere synthesis in ALT+ cancer cells [4]. BIR is initiated by the resection of DSB ends that have invaded a homologous template donor, followed by DNA synthesis. The stalled replication forks at ALT telomeres can perform a 5′ end excision by recruiting BLM helicase, followed by exonucleases (e.g., EXO1 and DNA2), to produce single-stranded telomeric DNA [71] and proceed with 5′-3′ end resection. The WRN helicase has also been shown to be associated with ALT telomeres and can perform 5′ end excision by interacting with DNA [72,73]. In addition, MRE11 interaction with RAD50 and NBS1 (MRN complex) can also mediate end resection [74] (Figure 2B).

In addition to HR between telomeres, ALT+ cells can initiate intratelomeric recombination through an unknown mechanism and thus form T-loops at the telomere ends (Figure 3A) [75]. The homologous reassembly of T-loops is blocked by TRF2, which recruits RTEL1 to unwrap the T-loops in the S phase, thus protecting the T-loops from excision by the structure-specific endonuclease subunit, SLX4 [76]. It was shown that the telomere localization of SLX4 is higher for damaged telomeres than undamaged ones, and the endonuclease activity of SLX4 is required for T-SCEs and telomere recombination in ALT cells [77]. The DSB undergoes rapid 5′-3′ end resection, and the 3′ overhang formed after resection is bound by RPA and can be repaired in three ways. First, when DSB occurs in non-telomeric regions, it can be repaired by alternative nonhomologous end joining (Alt-NHEJ), mediated by PARP1 and DNA polymerase Polθ. This process requires LIG3 [76,78] (Figure 3B), which may inhibit the ALT pathway by repairing the internal telomeric DSBs before HR. Second, the overhang can be repaired by intratelomeric recombination, mediated by RAD51 and RAD52. This recombination process requires RAD51-mediated chain invasion [15] (Figure 3C). It has been shown that in cells with an overexpression of the TRF1-Fok1 fusion protein, telomeric DNA end excision, telomere aggregation in APBs, C-circle generation, and telomere-associated DNA synthesis is evident and the exposed telomere ends are bound by RPA and RAD51 [79]. Moreover, the RAD51-bound telomeres in TRF1-Fok1-overexpressing cells invade other telomeres and induce telomere aggregation. Furthermore, HOP2-MND1 also binds to damaged telomeres and is required for RAD51-mediated chain invasion [80]. The proportion of chain invasion and telomere aggregation is greatly decreased in RAD51-deficient cells. The deletion of HOP2–MND1 also decreases the telomere aggregation in APBs [80]. Finally, the telomere terminal 3′ overhang can be repaired by an unknown mechanism involving intratelomeric recombination [81]. And the intratelomeric recombination doesn’t require the involvement of RAD51 and HOP2-MND1 [11]. (Figure 3D).

## 5. Telomere Extension in ALT: Two Distinct Break-Induced Replication Pathways

Telomere elongation through ALT depends on homology-directed repair. However, recent studies with mammalian cells have demonstrated that ALT is a BIR-related process. Research has shown that BIR of ALT telomeres can proceed via two distinct mechanisms that facilitate telomere elongation: RAD51-dependent and RAD52-dependent processes [15].

RAD51 promotes the search for, and capture of, homologous template DNA during recombination and possibly stabilizes replication forks when undergoing replication stress [82]. Previous studies have shown that the initiation of ALT requires RAD51-mediated chain invasion and that this process requires HOP2-MND1 participation [80]. The telomeric DNA 3′-overhang, formed after enzymatic cleavage, is bound by RPA to protect the single-strand DNA (ssDNA) overhang. This is immediately followed by RAD51 substitution for the homologous chain search and strand invasion to form a D-loop [15,80]. However, a different study indicated that the deletion of RAD51 and HOP2, after telomeric DNA damage response (DDR) induced by TRF1-FokI, did not affect BIR, but still stimulated the synthesis of telomeric DNA [79]. These findings suggest that another BIR mechanism, that does not require a homology search and is RAD51-independent, is involved (Figure 4A).

Zhang et al. demonstrated that RAD52 is essential for APB generation and ALT-maintained telomeres [15]. RAD52 is also important for DNA DSB repair and HR [11]. 5-Ethynyl-2′-deoxyuridine (EdU) is a thymidine analog that can be incorporated into replicating DNA for the detection of cell proliferation. It has been shown that the number of APBs and EdU+ APBs decreased significantly in RAD52-deficient ALT+ cells, indicating that RAD52 is necessary for ALT telomere synthesis [15]. RAD52 can bind to DNA ends and mediate the DNA interactions required for DNA strand annealing [83]. Evidence has suggested that RAD52 can promote the invasion of ssDNA into double-stranded telomeric DNA in the presence of RPA, thereby promoting the generation of a D-loop (Figure 4A). However, the mechanism is unclear.

Are these two pathways equally important in ALT+ cells? Research has suggested that, in RAD52-knockout cells, telomeres were gradually shortened, whereas C-circle formation and ALT-DNA synthesis were increased. The absence of RAD52 expression led to the progressive activation of another ALT pathway that led to C-circle production and the gradual shortening of telomere length, which could not fully compensate for the loss of RAD52 [25]. In the presence or absence of RAD52, the absence of RAD51 did not significantly reduce telomere DNA synthesis, which demonstrates that RAD51 was not a major contributor in either of the two ALT pathways. However, the loss of RAD51 led to elevated C-circle formation, which confirms that RAD51 inhibited the formation of C-circles through an RAD52-independent pathway [15].

Both the RAD51- and RAD52-dependent processes of search and capture promote the formation of joined DNA structures called D-loops (Figure 4B). The D-loop structure is a platform for recruiting DNA polymerase and initiating DNA synthesis [11]. ALT telomere synthesis depends on the recruitment of the DNA polymerases Polδ and Polη [11]. Polη has been previously shown to extend the D-loop in vitro, suggesting that Polη may have the potential to initiate or trigger DNA synthesis [84]. ALT-mediated break-induced telomeric DNA synthesis is driven by a replisome consisting of POLD3-PCNA-RFC1. Furthermore, PCNA and POLD3 have been found to be recruited to damage sites before the occurrence of telomeric DSBs, suggesting that PCNA and RFC1 are damage sensors through which the replisome is activated [85].

After DNA synthesis begins, the POLD3 subunit of Polδ replaces Polη, and longer DNA fragments are synthesized to facilitate telomere extension [69]. POLD3 is required to stabilize the Polδ complex, and it interacts with PCNA during DNA synthesis and strand displacement [69]. Multiple lines of evidence have indicated that POLD3 is a major component of the replisome and is required for break-induced telomeric DNA synthesis [69,86].

## 6. Unraveling Recombinant Intermediates in ALT+ Cells

At the conclusion of DNA synthesis, recombination intermediates at ALT telomeres must be processed before mitosis to ensure they dissociate from the intertwined DNA molecules and thus maintain chromosome stability [87]. The recombinant intermediates produced by ALT telomeres can be processed mainly in two ways. First, they can be directly cleaved by structure-specific nucleases, e.g., the SMX complex (SLX1-SLX4, MUS81-EME1, and XPF-ERCC1), which eventually produces telomeres with exchange sequences [88].

In human cells, the molecular scaffolding protein SLX4 recruits the structure-specific nucleic acid endonucleases, SLX1-MUS81-EME11 and ERCC1/ERCC4, to a DSB repair site where these complexes cleave the D-loop [83] (Figure 4D). The dissociation induced by SLX4 depends on the related exonuclease. Through its nuclease activity, the SMX complex cleaves a DNA strand, which unravels the recombination intermediates, leading to telomere exchange events without telomere extension [9]. A study by Sobinoff et al. suggested that overexpression of SLX4 increases the expression level of SLX1 and the frequency of T-SCEs, in ALT cells [11]. Telomere synthesis was counteracted by the SLX4-SLX1-ERCC4 complex, which inhibits the association of POLD3 and ALT telomeres, and promotes the dissociation of telomeric recombinants [9].

Second, the resolution of recombination intermediates formed at ALT telomeres proceeds via the migration of crossed DNA strands in the presence of the dissociation complex BTR (containing BLM, TOP3α, RMI1, and RMI2) and Rad54 (Figure 4C), which eventually produces telomeric sequences without exchange [9]. Through its ATPase activity in vitro, RAD54 can both bind a D-loop and promote branch migration, which indicates that RAD54 can promote ALT. The deletion of RAD54 also leads to the formation of unresolved intermediates at telomeres, which form ultrafine anaphase bridges during mitosis [88]. The dissolution of D-loops is regulated by the RecQ helicase BLM, which forms a scaffold for TOP3α, RMI1, and RMI2 to generate the BTR isomerization complex (Figure 4C) [9,89]. The BTR complex is required for ALT-mediated telomere synthesis [66]. The recombinant intermediates formed during the chain invasion are processed through the BTR complex, initiating rapid POLD3-dependent telomere synthesis and then dissolution without the exchange of telomeric DNA [11]. It has been revealed that the deletion of any protein in the BTR complex in ALT+ cells causes an increase in C-circle abundance and T-SCEs. Moreover, the deletion of the BTR complex also leads to a significant reduction in telomere length in ALT+ cells [89]. This process is counteracted by the SLX4-SLX1-ERCC4 complex. The BTR complex promotes the dissolution of the recombination intermediates and thus promotes telomere extension in the absence of telomere exchange. This outcome illustrates that the BTR complex promotes ALT-mediated telomere synthesis and subsequently promotes telomere dissolution, and that BLM requires the Polδ components, POLD3 and RAD51, to initiate these events. These results indicate that SLX4 inhibits ALT telomeric DNA synthesis by cleaving the recombinant intermediates formed during BIR, whereas BLM carries out telomeric DNA lengthening through the promotion of POLD3-mediated telomeric DNA synthesis and the dissolution of recombinant intermediates [88].

BLM helicase activity inhibits SLX4 nuclease activity during telomeric D-loop and HJ formation in vitro [89]. Furthermore, BLM inhibits the formation of telomeric loops and T-SCEs modulated by SLX4 when induced in vivo [9]. Both BLM and SLX4 preferentially localize to the telomeric DNA associated with APBs in ALT+ cells. The combined deletion of BLM and SLX4 can lead to a significant reduction in C-circle formation and T-SCE rates, reduced telomere length, and the slight inhibition of telomere extension in ALT cells [90]. These findings indicate that both SMX and BTR complexes are required in ALT+ cells, and that ALT pathway activity is the balance of dissolution and resolution of recombination intermediates [9].

## 7. Conclusions

### Targeting ALT-Related Cancers

ALT maintains telomere length and genome stability in ALT+ cells. ALT-related tumors are highly heterogeneous, and the ALT phenotype is relatively common in certain sarcoma and astrocytoma subtypes [91]. However, ALT activity is nearly undetectable in normal tissues or benign tumors. Therefore, the ALT mechanism shows great potential as a target for the treatment of telomerase-negative tumors. The previously mentioned studies not only have implications for understanding the molecular basis of the ALT pathway, but also for developing therapeutic strategies for ALT tumors.

In recent decades, several studies have focused on ALT-related tumor-targeting drugs. For example, Lee Zou’s group demonstrated that VE-821, an ATR inhibitor, shows excellent targeting of ALT+ cells [92]. Research by Yong Zhao’s team showed that the cisplatin derivative, Tetra-Pt (bpy), stabilizes the secondary structure of G-quadruplexes on DNA, thereby inhibiting the telomeric HR rate to specifically kill ALT+ tumor cells [93]. In addition, Luca Pompili et al. showed that ALT+ cells are sensitive to treatment with trabectedin [94]. Joydeep Mukherjee et al. demonstrated that PARP inhibitors such as olaparib involve the stimulation of c-NHEJ-dependent (classical non-homologous end joining) telomere fusion and selectively kill ALT+ tumor cells [95].

Considering previous research on the ALT pathway mechanism, we summarize the potential therapeutic targets of ALT as follows. Firstly, the activity of ALT+ cells can be inhibited by suppressing HR through, for example, ATR inhibitors and G-quadruplex stabilizers. Secondly, ALT pathway maintenance depends on the balance between DNA damage and repair; therefore, the death of ALT+ cells can be induced by altering the homeostasis of ALT telomeres. Finally, the proliferation of ALT-related tumor cells can be inhibited by inducing telomere fusion in ALT+ cells. PARP has been found to bind specifically to ALT telomeres and is involved in stabilizing TRF2 binding and preventing lethal telomere fusion [95]. Therefore, inhibiting the activity of PARP or the proteins involved in the inhibition of ALT telomere fusion is a strategy for selectively killing ALT-related tumors. A growing number of experiments have demonstrated that ALT is a finely tuned system that maintains telomere length in a recombination-dependent manner. Further understanding of ALT regulatory mechanisms will provide new insights into targeted therapy for ALT-related tumors.

## Figures and Tables

**Figure 1 cancers-14-02194-f001:**
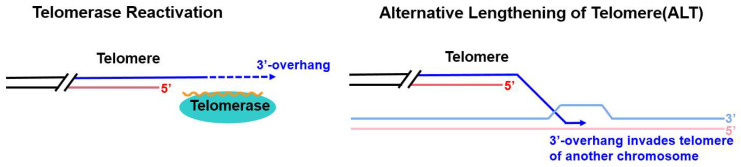
Two distinct telomere extension mechanisms. Left: Through reverse transcriptase activity, telomerase extends the 3′-overhang of telomeres. Right: ALT pathway action depends on homologous recombination (HR) for telomere extension and break-induced replication (BIR), in which the 3′ DNA overhang invades homologous double-stranded telomeric DNA of another chromosome, leading to the formation of a DNA displacement loop (D-loop). The invading strand then serves as the primer for the initiation of DNA replication.

**Figure 2 cancers-14-02194-f002:**
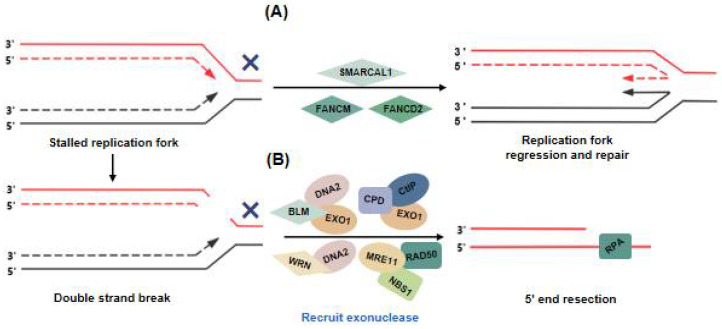
The processing of stalled replication fork at ALT telomeres. (**A**) SMARCAL1, FANCD2 and FANCM protect stalled forks from collapsing and promote replication fork regression and repair. (**B**) The DSBs generated by fork collapse undergo end resection, which is mediated by BLM -DNA2-EXO1, CPD-CtIP-EXO1, WRN-DNA2, or the MRE11-RAD50-NSB1 complex.

**Figure 3 cancers-14-02194-f003:**
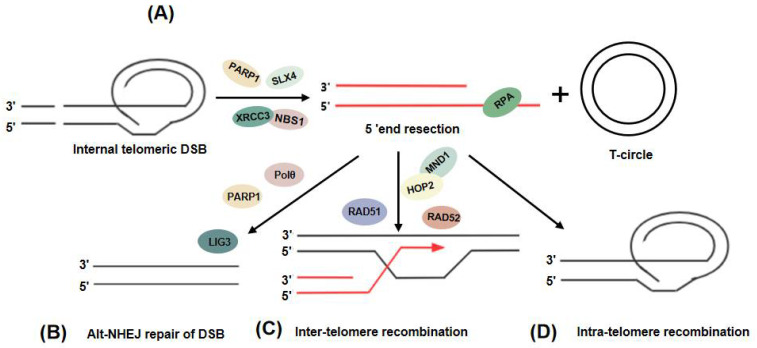
Internal telomeric double-strand breaks (DSBs) result in intratelomere recombination and the creation of T-circles in ALT cells. (**A**) Internal telomeric DSB recruitment of nucleases results in 5′ end cleavage and the creation of a T-circle. (**B**) Alternative nonhomologous end joining (Alt-NHEJ) repair of DSBs. (**C**,**D**) The 3′ overhang allows intertelomere recombination and intratelomere recombination.

**Figure 4 cancers-14-02194-f004:**
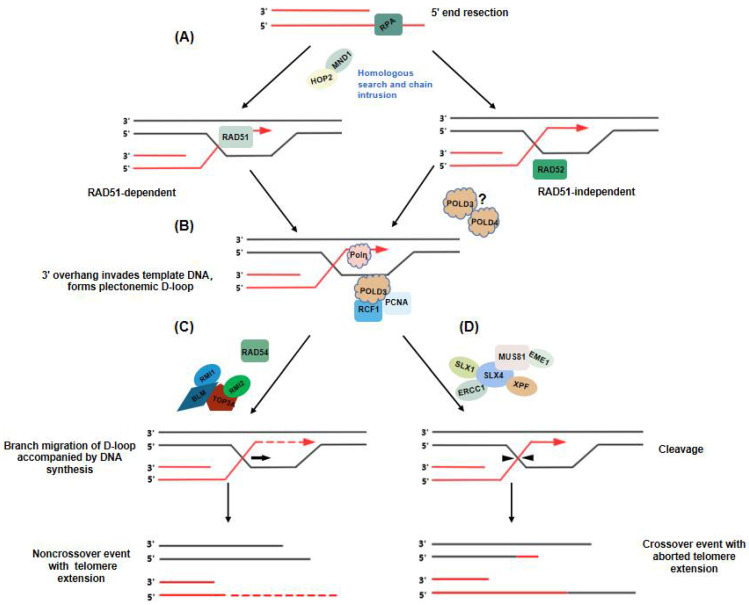
D-loop migration and unraveling of recombinant intermediates. (**A**) Homologous search and chain invasion mediated by HOP2–MND1. (**B**) Telomeric DNA synthesis, promoted by DNA polymerase. (**C**) D-loop migration, promoted by RAD54. RAD54 may function in coordination with BLM to promote BTR complex-dependent dissolution of recombination intermediates. (**D**) Intermediates unraveling can also be achieved through SMX complex recruitment, leading to telomere exchange without telomere extension.

## Data Availability

Not applicable.

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
