# Peer review of "Alternative Lengthening of Telomeres and Mediated Telomere Synthesis"

_cancers, 2022, doi:10.3390/cancers14092194_

Round 1
Reviewer 1 Report
This is an excellent review with lots of molecular details of the ALT mechanism. It seems factual correct and is of high scientific standard. I have just very few rather minor comments that have to be addressed and I list below:
- Please also add that in addition to some cancers (mainly sarcomas) ALT also frequently occurs during experimental cellular immortalisation of mesenchymal cells.
- page 1, 1st paragraph of intro: Please specify that you refer to embryonic stem cells (ESCs) since adult/tissue stem cells are mainly quiescent and only activate TA when activated and progenitor cells also shorten their telomeres.
- In the introduction, 2nd paragraph: Not just telomerase activity is responsible for immortalisation-it just extends cellular lifespan, but the events responsible for immortalisation (oncogene activation and inhibition of tumour suppressors) occur before TA activation and are just conserved due to telomere maintenance and extending proliferation capability.
- The telomerase RNA component (451nt) is not identical with the template (just 11nt) but contains it.
- page 2 under point 1: Please add the end replication problem (ERP) as well as oxidative stress as the main causes for telomere shortening rather than just cell division.
- Page 2: in my view telomere length (TL) is rather heterogenous (from very long to very short) rather than "uneven" while I am not so sure about the heterogeneity of the cells other than regarding to TL. Perhaps specify further what you mean here for the cells.
- page 3: instead of "chains" better use the term "strands" when referring to DNA.
- At the end of page 3: what do you mean with "structures detected in telomeres"? There are just repeated hexanucleotides, or do you include subtelomeres? please specify.
- page 7 regarding a repair of the 3' ss overhang: any break would be a ss break leading to loss of DNA sequence. As the length of the overhang is known to vary-are you sure that there is a repair possible? Please provide a reference for that.
- page 8: As mentioned above, 3' ss overhangs are formed due to the ERP and bound mainly by Pot1 or do you refer to the 5' overhang? Please specify.
- page 10 end of 1st paragraph: What exactly is "telomere splitting"? Please explain.
- page 11, 2nd paragraph: There seems to be a contradiction since in 1 sentence you describe the induction of telomere fusion as a therapeutic target while in the next sentence you recommend parp that inhibits telomere fusion as a potential therapy. Please reconcile these contradicting statements.
minor language and formal mistakes: Please leave a space before the reference citation at the end of a sentence. On page 1: remove "the" in front of "replicative" and "genomic". Page 5 under "2" line 4: remove "a" in front of "many", under 2.1 line 10 remove "to" in front of "on". When referring to specific groups, please only use the surname of the scientists.. Please explain that FA means Fanconi. page 6 under 3: line 5: add "of" between recruitment and "specific", page 7, end of 1st paragraph: "bindS to...", figure legend 3: it should be rather "end joining" than "joint", page 7, 3rd line from below: "ALT depends", remove "has", page 8, 3rd paragraph: please explain "Edu" and correct "Rad52 can bind to a DNA end" or "to DNA ends", 2nd last line of page 8: instead of "proves" better say "confirms", page 9: instead of pold it should rather be delta. with "before telomeric DSBs"-do you mean that spatially or time-wise? When counting, say "firstly, secondly...", pre-last line on page 10 it should be "balance" as only used in singular. page 11, 4th line 2nd paragraph: "on THE balance". Please order abbreviations by the alphabet
Author Response
- Please also add that in addition to some cancers (mainly sarcomas) ALT also frequently occurs during experimental cellular immortalisation of mesenchymal cells.
Response: We have added this sentence in the abstract and marked it by red.
- page 1, 1st paragraph of intro: Please specify that you refer to embryonic stem cells (ESCs) since adult/tissue stem cells are mainly quiescent and only activate TA when activated and progenitor cells also shorten their telomeres.
Response: As suggested by the reviewer, we have added following sentences in the introduction and marked them by red.
Except in embryonic germ cells, stem cells, and cancer cells, telomere length gradually shortens with cell division.
- In the introduction, 2nd paragraph: Not just telomerase activity is responsible for immortalisation-it just extends cellular lifespan, but the events responsible for immortalisation (oncogene activation and inhibition of tumour suppressors) occur before TA activation and are just conserved due to telomere maintenance and extending proliferation capability.
Response: We have already edited and marked it by red.
To escape from the “Hayflick limit”, the majority of tumor cells reactivate telomerase, which maintains telomere length.
- The telomerase RNA component (451nt) is not identical with the template (just 11nt) but contains it.
Response: Thanks for pointing out this mistake we made, and the sentence which descripts telomerase RNA component has been corrected and marked by red.
- page 2 under point 1: Please add the end replication problem (ERP) as well as oxidative stress as the main causes for telomere shortening rather than just cell division.
Response: As suggested by the reviewer, “…the shortening of telomeres caused by continuous cell division as well as oxidative stress” was added and marked by red.
- Page 2: in my view telomere length (TL) is rather heterogenous (from very long to very short) rather than "uneven" while I am not so sure about the heterogeneity of the cells other than regarding to TL. Perhaps specify further what you mean here for the cells.
Response: We agree that heterogenous is better in descriping telomere length (TL) than uneven. And this word is corrected and marked by red.
- page 3: instead of "chains" better use the term "strands" when referring to DNA.
Response: We have replaced the “chains” by “strands” when referring to DNA and marked by red.
- At the end of page 3: what do you mean with "structures detected in telomeres"? There are just repeated hexanucleotides, or do you include subtelomeres? please specify.
Response: We have specified the "structures detected in telomeres" by “…which are the majority of extrachromosomal telomeric circle structures detected in the ALT cells and function as substrates for the generation of extrachromosomal telomeric circles, such as C-circles”.
- page 7 regarding a repair of the 3' ss overhang: any break would be a ss break leading to loss of DNA sequence. As the length of the overhang is known to vary-are you sure that there is a repair possible? Please provide a reference for that.
Response: In this part, we were trying to explain how the internal telomeric DSBs processing at ALT telomeres. We’ve already made it more explicit.
- page 8: As mentioned above, 3' ss overhangs are formed due to the ERP and bound mainly by Pot1 or do you refer to the 5' overhang? Please specify.
Response: The single stranded 3’-overhang is induced by exonuclease resection, and the 3’ tails are coated by ss DNA-binding protein RPA as soon as they are generated. We’ve already specified it.
- page 10 end of 1st paragraph: What exactly is "telomere splitting"? Please explain.
Response: Here we want to express the meaning of the dissociation of telomeric recombinants. We have already edited and marked this part by red.
Telomere synthesis was counteracted by the SLX4-SLX1-ERCC4 complex, which inhibits the association of POLD3 and ALT telomeres and promotes the dissociation of telomeric recombinants.
- page 11, 2nd paragraph: There seems to be a contradiction since in 1 sentence you describe the induction of telomere fusion as a therapeutic target while in the next sentence you recommend parp that inhibits telomere fusion as a potential therapy. Please reconcile these contradicting statements.
Response: Sorry for the unclear expression and we have already edited this part and marked it by red.
Inhibiting the activity of PARP or proteins involved in the inhibition of ALT telomere fusion is a strategy for selectively killing ALT-related tumors
minor language and formal mistakes: Please leave a space before the reference citation at the end of a sentence. On page 1: remove "the" in front of "replicative" and "genomic". Page 5 under "2" line 4: remove "a" in front of "many", under 2.1 line 10 remove "to" in front of "on". When referring to specific groups, please only use the surname of the scientists.. Please explain that FA means Fanconi. page 6 under 3: line 5: add "of" between recruitment and "specific", page 7, end of 1st paragraph: "bindS to...", figure legend 3: it should be rather "end joining" than "joint", page 7, 3rd line from below: "ALT depends", remove "has", page 8, 3rd paragraph: please explain "Edu" and correct "Rad52 can bind to a DNA end" or "to DNA ends", 2nd last line of page 8: instead of "proves" better say "confirms", page 9: instead of pold it should rather be delta. with "before telomeric DSBs"-do you mean that spatially or time-wise? When counting, say "firstly, secondly...", pre-last line on page 10 it should be "balance" as only used in singular. page 11, 4th line 2nd paragraph: "on THE balance". Please order abbreviations by the alphabet
Response: All of them are already edited and marked by red

Reviewer 2 Report
Currently, there are many similar review articles on ALT mechanisms. In my opinion, the authors of the manuscript were able to concisely describe a summary of the information known so far on ALT. Schematic representations could provide more information and be of better quality.
I have no other serious comments.
Minor comments
The abbreviations APB and PARP should be included in the list of abbreviations. Abbreviations should be arranged alphabetically.
The spaces between the end of the sentence and the quote are very often missing.
Especially the legend of Fig. 2 is too brief. Since figure is too schematic, it would be appropriate to at least give more details in the description so that it is legible at first sight.
Author Response
Minor comments
The abbreviations APB and PARP should be included in the list of abbreviations. Abbreviations should be arranged alphabetically.
Response: Thanks for the reminder, we have corrected it.
The spaces between the end of the sentence and the quote are very often missing.
Response: Thanks for the reminder, we have corrected it.
Especially the legend of Fig. 2 is too brief. Since figure is too schematic, it would be appropriate to at least give more details in the description so that it is legible at first sight.
Response: We have already edited and marked it by red.
